# Nuclear-Localized Fluorescent Proteins Enable Visualization of Nuclear Behavior in the Basidiomycete *Schizophyllum commune* Early Mating Interactions

**DOI:** 10.3390/jof9111043

**Published:** 2023-10-24

**Authors:** Marjatta Raudaskoski, Ciarán Butler-Hallissey

**Affiliations:** 1Molecular Plant Biology, Department of Life Technologies, University of Turku, FIN-20014 Turku, Finland; 2Turku Bioscience Centre, University of Turku and Åbo Akademi University, FIN-20014 Turku, Finland; ciaran.butler-hallissey@univ-amu.fr; 3Aix-Marseille Université, CNRS, INP UMR7051, NeuroCyto, 13005 Marseille, France

**Keywords:** cell cycle, filamentous basidiomycete, nuclear fluorescence, histone replication, mating-type genes

## Abstract

Spinning disc confocal microscopical research was conducted on living mating hyphae of the tetrapolar basidiomycete *Schizophyllum commune*. Haploid strains with either the same or different *A* and *B* mating-type genes and expressing differently labelled histone 2B were confronted. In the haploid hyphae histone 2B mCherry and histone 2B EGFP were visualized as red and green nuclei, respectively. In hyphae with the same *A* but different *B* genes, the red and green nuclei were observed next to each other. This indicated that nuclear migration between strains, regulated by the *B* mating type, had taken place. The compatible mating with different *A* and *B* genes produced a high number of mixed EFGP/mCherry, yellow nuclei. The mixed nuclei resulted from nearby divisions of nuclei encoding different histones and mating-type genes. During this process, the histones with the different labels were incorporated in the same nuclei, along with the heterodimerized transcription factors encoded by the different *A* mating-type genes and present around the nuclei. This led to the activation of the A-regulated pathway and indicated that different *A* genes are important to the cell cycle activation of a compatible mating. Consequently, a yellow nuclear pair stuck together, divided synchronously and proceeded in the migration hyphae towards the colony periphery, where the dikaryotization was promoted by branch formation from the migration hyphae.

## 1. Introduction

The filamentous basidiomycetes in the class Agaricomycetes are saprotrophic, wood degrading, plant pathogenic and mycorrhizal fungi best recognized from their fruiting bodies, which develop from an extensive dikaryotic mycelium. The growth of the dikaryotic mycelium is dependent on the extension of the hyphal tip cells and the synchronous (conjugate) division of the two nuclei that are different in their mating-type genes in the hyphal tip cells. In most Agaricomycetes, clamp cell development is associated with the synchronous nuclear division. The mating-type genes regulating the dikaryotic growth, consist of two unlinked complexes A and B. Each complex consists of two loci each, with *Aα* and *Aβ* and *Bα* and *Bβ* genes. The formation of the dikaryotic mycelium starts from the fusion of haploid homokaryotic hyphae with different *A* and *B* mating-type genes, either having different genes at *Aα* and/or *Aβ* and at *Bα* and/or *Bβ*. The *A* and *B* genes are also called *HD* and *PR* mating-type genes [1]. The *B/PR* genes consist of genes encoding pheromones and a G-protein coupled receptor (GPGR) and the A/HD genes are homeodomain transcription factors. Sexual reproduction takes place if the mated haploid hyphae carry different genes for the pheromones, and receptors and homeodomain genes. During the development of dikaryotic hyphae, reciprocal exchange and migration requires different *B* mating-type genes, while the nuclear pairing and synchronous division requires different *A* mating-type genes. The fusion of the clamp cell tip with the subapical cell and the migration of the nucleus into the subapical cell requires different *B* genes [1,2,3,4,5,6].

The synchronous nuclear division (conjugate division) regulated by *A* and *B* mating-type genes in dikaryotic hyphae has been visualized in both fixed and living hyphae [7,8,9,10]. At division, one of the two nuclei divides in the hypha and the other one at the base of the developing clamp connection. The sister nuclei from the division in the hypha are quickly separated, one to the apical and the other to the subapical part of the apical cell. One of the sister nuclei from the division at the base of the growing clamp cell also proceeds towards the hyphal apex, while the other remains enclosed in the clamp cell due to the formation of septa in the hypha and at the base of the clamp cell. The nucleus is released after the fusion of the clamp cell tip with the subapical cell, and it then moves next to the nucleus in the subapical cell.

The molecular structure and the pathways regulated by the *A* and *B* genes during the mating process and dikaryotic growth are well-known in the model fungi *Schizophyllum commune* [2,11] and *Coprinopsis cinerea* [3], in the basidiomycete smut *Ustilago maydis* [12] and in several other filamentous basidiomycetes as well [1,13,14]. Instead, the downstream targets (activity) of either the *A* or *B* genes during the dikaryotization process between haploid compatible strains are not yet understood. In the present study, the strains of *S. commune* carrying different *A* and *B* mating-type genes and nuclei with different histone 2B fluorescent tags, mCherry and EGFP, respectively, were used to observe the mating interactions in living hyphae. During mating, the hyphae grow into the opposite colony and hyphal fusions take place. With a widefield fluorescence microscope, or even point scanning with confocal microscopy, it is difficult to follow the fate of the nuclei in the hyphal colony due to the fusion and intermingling of the hyphae in a colony that is often several layers thick. Therefore, scanning the mated colonies with differently labeled nuclei using spinning disc confocal microscopy provided new information about the nuclear behavior in the establishment of a dikaryon from compatible haploid strains of *Schizophyllum commune.*

In the confronted haploid strains, the different *B* mating-type genes induce the reciprocal exchange and migration of nuclei as well as the breakdown of the septa in the interacting strains [2]. This is seen as a “flat” phenotype, not recognized in other filamentous basidiomycetes. Microscopically, the “flat” phenotype has mainly been studied in the haploid *B*-mutant strain, which is equivalent to mating with different B genes (A = B ≠) [15,16], and to a lesser extent in the confrontation of haploid strains with common *A* but different *B* genes, as in the present study using spinning disc confocal microscopy. A compatible mating is usually recognized by the formation of clamp connections at the colony edge. No information is available about the *A*-regulated process that organizes the nuclei into synchronously dividing nuclear pairs with different mating-type genes. Analyses of the migration hyphae of a compatible mating, in which the nuclei with different mating-type genes are recognized due to their fluorescent colours, revealed that nuclear divisions play a central role in the dikaryotizaton process.

## 2. Materials and Methods

### 2.1. Culture Conditions, Strains, Cloning and Mating Interactions

*Schizophyllum commune* strains were grown on complete medium [17] but supplemented with 1% glucose. All the haploid strains used in the present experiments are stored in complete medium at 4 °C in darkness. At matings, the proper haploid strains were inoculated opposite each other, 0.5 cm apart, on cellophane membranes overlaying complete medium and covered with a thin layer of 0.5% agarose in complete medium. After 24 h of growth at 29 °C in darkness, the colonies were examined under a dissecting microscope, and the matings with the initial touch of the hyphae from different strains were selected for spinning disc confocal microscopy. To observe later hyphal interactions, the growth of the cultures was then continued further for 4 to 12 h. The center of these older cultures was examined with spinning disc confocal microscopy for nuclear structure in mycelium consisting of fused and intermingling hyphae from both strains. The periphery of the cultures was also screened for the development of dikaryotic hyphae.

Among haploid strains employed, I-IIH2B::EGFP (*A26Bα4-β1 ura- h2B::egfp phleom+*), 4 H2B::EGFP (*A43Bα3-β6 h2B::egfp phloem+*), and F16 H2B::EGFP (*A43Bα4-β1 h2B::egfp phloem+*) were already designed for the first fluorescence microscopy imaging of the living hyphae in *S. commune* [9]. The strains were also used in the study of the relationship between the nuclear division and synthesis of septa [10]. The strain T14-3-7 (*A43Bα3-β6 h2B::mCherry phleom+*) expressing H2B::mCherry was constructed for the present work.

The primers for mCherry (Clonetech, CatNo 632542) cloning were mCherryF GGGGATCCACCGGTCGCCACCATGGTGAGCAAGGG and mCherryR GCGCGGCCGCGAGACTAGTTTCCGGACTTG, with *Bam*H1 and *Not*I restriction sites. The amplified PCR product, containing a linker of seven amino acids (G, D, P, P, V, A, T) in front of mCherry was cloned into the pCR2.1 TOPO vector (Invitrogen). The histone *h2B* gene with an 826 bp native promoter was released from plasmid pN-Hist2B::EGFP [9] by *Bam*H1 and cloned in the right direction in a mCherry-containing plasmid opened with *Bam*H1. The plasmid containing pN-Hist2B::mCherry was named plasmid 14, a short terminator sequence from the gene *cdc42* was cut with *Not*I and cloned at the *Not*I site, and the phleomycin cassette was cloned at the *Apa*I site in plasmid 14 [18]. Plasmid 14, linearized with *Sca*I, was transformed into protoplasts from the haploid strain 1792-114-10 (*A43/Bα3-β6*) and the selection was made on complete medium containing 20 µg mL^−1^ phleomycin. The transformants were screened for mCherry fluorescence in nuclei and several transformants were obtained, out of which the T14-3-7::mCherry transformant was used in the present work.

Spinning disc confocal microscopy was performed on three types of matings between haploid strains (Table 1). (1) In the mating T14-3-7::mCherry X 4H2b::EGFP, the *A* and *B* mating-type genes are the same (Table 1, A = B =), only the fluorescence of the nuclei is different. This is an incompatible mating. No reaction apart from hyphal fusions is expected to occur between the strains. (2) In the mating interaction I-IIH2B::EGFP X T14-3-7H2B::mCherry, the haploid strains have different *A* and *B* mating-type genes (Table 1, A ≠ B ≠), and the *h2B* gene is labeled with a different fluorescent marker. Due to different *A* and *B* genes, a compatible mating leads to the formation of dikaryotic mycelium. (3) Mating of strain T14-3-7H2B::mCherry with strain F16 H2B::EGFP is a hemicompatible mating, with the same *A* gene but with different *B* genes (Table 1, A = B ≠), leading to a “flat” mycelial morphology due to the activity of the different *B* mating-type genes inducing reciprocal nuclear migration between the strains and septal dissolution in the strains. In mating with different *A* but similar *B* mating-type genes (A ≠ B =), no extensive reciprocal exchange and migration of nuclei take place, and it is not examined here.

### 2.2. Spinning Disc Confocal Microscop

For spinning disc confocal microscopy, the thin layer of agarose with mycelium was separated from the cellophane membrane in distilled water and transferred to a glass bottom culture dish (Mat Tek Corporation pp35G-1.5-14-C, Ashland, MA, USA). Images were collected with a 3i Marianas CSU-W1 spinning disc (Intelligent Imaging Innovations, Inc., Göttingen, Germany) (50 µm pinholes) confocal microscope, equipped with an inverted Zeiss Axio Observer (Camera: Teledyne IImaging Imaging, Tuscon, Arizona, USA) 7 microscope and 40× Zeiss LD objective N.A 0.6 and 63× Zeiss LD Plan-NEOFLUAR objective N.A 0.75. For fluorescence images, EGFP was excited with a 488 and mCherry with a 561 nm solid-state diode laser. The images were recorded with a Photometrics Prime BSI sCMOS camera (Teledyne Imaging, Tuscon, Arizona, USA). Either a single-image plane or a stack with 1.25 µm intervals was obtained with the fluorescence and bright field channels. The images were mainly captured with the 40× objective, unless otherwise stated, and further processed with Image J software (1.53q) [19]. For maximum Z projections, a specific area and number of Z slices were selected. The 3D images were acquired by reconstructing maximum Z projections of the XY sections. Image J color split channels and Z-plot analysis were used to examine the expression of the fluorescent proteins in nuclei. The brightness, contrast and size of the digital images were optimized using Corel PHOTO-PAINT 2021 (64-Bit), CORELDRAW 2021(64-Bit).

### 2.3. Recording of Nuclear Numbers with Different Colors

For recording of red, green and yellow nuclei, mating interactions of the same age were chosen. For incompatible and compatible mating interactions, only one sample from each with the same mating duration (about 36 h) was available. For the hemicompatible, common-A (A = B ≠) mating two samples of the same age were examined. Counting of nuclei expressing EGFP, mCherry and both EGFP/mCherry was performed manually from Z-projections of nine slices, 29,071 × 29,071 μm in the area, with red, green and brightfield images (Table 2).

## 3. Results

### 3.1. Spinning Disc Confocal Projections of Mating Interactions with Different-Colored Nuclei—An Overview

In the incompatible mating (A = B =), in which the strains have the same *A* and *B* genes (Table 1) but differently labelled nuclei, the nuclei are in separate hyphae and have an elongated shape even when the hyphae from both strains are intermingled (Figure 1a,b). In a compatible mating (A ≠ B ≠), by strains with different *A* and *B* genes (Table 1) and of the same age as in the previous mating (Figure 1a,b), yellow and orange nuclei are observed in addition to the red and green ones (Figure 1c,d). The yellow and orange nuclei result from mCherry-labeled histone H2B accumulation into the nuclei with histone H2B׃׃EGFP and from histone H2B׃׃EGFP moving into the mCherry-labeled nuclei. The occurrence of yellow and orange nuclei indicates that nuclei with different mating types and colors have moved through hyphal fusions into the same hypha and lay in close proximity, leading to the incorporation of H2B::mCherry and H2B::EGFP into the same nucleus (Figure 1c,d). The fluorescence measuring of the H2B::mCherry and H2B::EGFP in the nuclei using Image J color split channels analysis suggested that in the yellow nuclei, the relative green and red fluorescence values in each nucleus were comparable (Figure 2). Further proof that both labeled histones are present in the same nucleus are obtained from the nuclear (conjugate) division in dikaryotic hyphae (Figure 1e,f), resulting from the mating of the same compatible strains as in Figure 1c,d. After synchronous nuclear division associated with clamp cell formation and telophase movement, except for the one remaining enclosed in the clamp cell, the nuclei increase their size and H2B::EGFP and H2B::mCherry are strongly expressed in each nucleus throughout the division, leading to the yellow nuclear color (Figure 1e and Appendix A).

On the other hand, in the visualization of nuclei in a common A (A = B ≠, Table 1), yellow nuclei are observed only occasionally, though the red and green nuclei occur next to each other in the same hypha (Figure 1g,h) indicating that the nuclear migration has taken place between the haploid strains but not the interaction between the nuclei expressing mCherry- and EGFP-labeled histones.

### 3.2. Details of Mating Interactions

In the haploid homokaryotic strains a green nucleus and a red nucleus are distinguished in the middle part of the apical cell, respectively (Figure 3a,b). The nuclear shape may vary from elongated ones or to small spheroids with tightly packed chromatin after nuclear division. When these strains with different *A* and *B* mating-type genes are inoculated opposite each other on thin culture medium, the hyphae grow past each other (Figure 3c) but also meet (Figure 3d). In the latter case, when a tip hits the side of another hypha, the tip branches strongly (Figure 3d and the enlargement). In this confrontation between compatible strains with differently colored nuclei, the nuclear number in both hyphae became irregular, indicating that nuclear divisions in both hyphae had taken place. The fusion at the site of the confrontation could not be confirmed (Figure 3d), but the fusion between hyphae would allow the nuclear movement with different mating-type genes into a branch, which could lead to the development of dikaryotic hyphae, which are often detected at the confrontation zone. In all mating interactions, fusions of hyphae from each strain were distinguished deeper in the opposed colonies. Fusions took place between hyphae with the same nuclear color but also between hyphae with red and green nuclei (Figure 3e–g).

The nuclear composition in migration hyphae is different in incompatible (A = B =), compatible (A ≠ B ≠) and hemicompatible, common-A (A = B ≠) matings (Figure 4, Table 2). Hyphal fusions, together with septal dissolutions and nuclear movements, lead to the development of clearly notable hyphae, the so-called migration hyphae with red and green nuclei next to each other, both in the compatible (A ≠ B ≠) and common-A (A = B ≠) matings. In the migration hyphae of the compatible (A ≠ B ≠) mating, the view of nuclei is vivid. In early mating, the migration hyphae contain sharply red and green nuclei without any color around them, but there are also nuclei surrounded by a dispersed yellow-green-red cloud (Figure 4a,b). The cloud is interpreted to represent the synthesis of histones around nuclei expressing the differently labelled histones. The cloud in Figure 4a,b vanished in two minutes, and after its disappearance, at the same site, four nuclei moving in the same direction were observed (Figure 4c, Appendix A). In the front, two closely associated nuclei occurred, green and yellow, followed by two separate nuclei, one reddish and the other greenish. When the synthesis of H2B::EGFP and H2B::mCherry occur in nuclei in close proximity, different amounts of H2B::EGFP and H2B::mCherry may accumulate into the nuclei.

The migration hyphae from a mating that was four hours older than in Figure 4a–c show multinucleate hyphae with red, green but also mixed EGFP and mCherry nuclei (Figure 4d), depending on the amount of green and red H2B included in the nucleus. The nuclei surrounded by histone clouds were also further recorded. The fluorescence measuring of the H2B::mCherry and H2B::EGFP in the nuclei (Figure 4d) using Image J color split channels analysis proved that in the yellow nuclei, the relative green and red fluorescence values were comparable as indicated by the previous measurements (Figure 1f and Figure 2). Both orange and yellow nuclei associated into pairs, which divided synchronously (Figure 4e). In a few hyphae, the multinucleate complex appeared to be surrounded by tracks passing through dissolved septa (Figure 4d, arrow).

In the common-A hyphae with only different B mating-type genes (A = B ≠), the migration hyphae contain red and green roundish nuclei pressed close to each other (Figure 4f), and yellow nuclei were observed only rarely (Figure 3g). The overall picture of the nuclei is stagnant. The narrow hyphae have long and short vacuolated compartments, often without nuclei. The hyphal septa are broken, and some hyphae are deformed.jof-09-01043-t002_Table 2Table 2The number of nuclei with green, red and yellow color at *Schizophyllum commune* different mating interactions of same age. Note the high number of yellow nuclei in the compatible mating. One sample (*) and two samples (**) with exactly the same age.Mating InteractionsNumber of Nuclei with Different Colors Total Number of Nuclei
GreenRedYellow
Incompatible A = B =4529175 *Compatible A ≠ B ≠1352572232 *Hemicompatible A = B ≠3459194 **

The nuclear colors were also recorded in full size images of incompatible (A = B =), compatible (A ≠ B ≠) and hemicompatible, common-A (A = B ≠) matings (Table 2). One yellow nucleus out of 75 nuclei was detected in the mating with the same *A* and *B* genes (A = B =) and one out of 94 nuclei in the mating with similar *A* but different *B* genes (A = B ≠). In the compatible mating (A ≠ B ≠) the total number of nuclei and the number of yellow nuclei was much higher at 232 and 72, respectively (Table 2). Image J color split channel analysis suggested that the few yellow nuclei detected in the incompatible and hemicompatible mating was not a result of overlapping nuclei, but that both H2B::mCherry and H2B::EGFP were expressed in the same nucleus. The occurrence of yellow nuclei, although in a low number, in the incompatible and hemicompatible matings requires attention in future research. 

### 3.3. Development of Dikaryotic Hyphae

In the periphery of an older colony with different *A* and *B* mating-type genes (A ≠ B ≠), the migration hyphae clearly contain nuclear pairs and clamp connection initials (Figure 5). Further analysis of EGFP and mCherry fluorescence indicated that each nucleus in the region contains both fluorescent histones exemplified by the insert in Figure 5. This proves that the nuclei originate from divisions that have taken place in close proximity in the central part of the colony. In the migration hyphae, the distance between the paired nuclei varies and nuclei move into the branches grown from the migration hyphae, exemplified by the description of three branches in Figure 5. A nuclear division in the branch may lead to the formation of a new branch (Figure 5, arrows with one or two stars). The detection of tip cells with two nuclei and a clamp connection (Figure 5, arrow heads and arrow with three stars) indicates that nuclei with different mating types have also moved from the migration hypha into a branch and a conjugate division has taken place. After the latter division, the apical cell of the branch is dikaryotic and contains nuclei with compatible *A* and *B* mating-type genes (Figure 5, arrow heads). The pairing of the nuclei with compatible mating-type genes in the migration hyphae, together with their movement into the branches grown from migration hyphae, leads to the development of a dikaryotic periphery in the colony. Following several migration hyphae to their very tip in bright field microscopy showed that the tip had an abnormal structure that suggested cessation of growth (not shown), probably due to the movement of the nuclei into the branches.

## 4. Discussion

The nuclear interactions are difficult to visualize in the early stage of a compatible mating leading to the formation of dikaryotic mycelium. In the present study, a clear difference in the nuclear distribution and composition was observed in the migration hyphae between incompatible (A = B =), hemicompatible, common-A (A = B ≠) and compatible (A ≠ B ≠) mating interactions expressing Histone-2B-EGFP (green) and Histone-2B-mCherry (red) fusion proteins. In the incompatible A = B = and common-A (A = B ≠) matings mainly red and green nuclei occurred, while in the mycelium with different *A* and *B* mating-type genes (A ≠ B ≠), a high number of yellow nuclei were seen in addition to the red and green ones.

The yellow nuclei were interpreted to result from the incorporation of histone H2B::mCherry and H2B::EGFP, both expressed under the *h2B* native promoter, into the same nucleus. In eukaryotic cells, including yeasts, DNA replication and histone synthesis are coupled to produce chromatin after nuclear division, at the early S phase of the cell cycle [20,21,22]. The core histone genes, such as *h2B*, are extensively but shortly transcribed at the beginning of the S phase [23,24]. In the migration hyphae of the compatible mating (A ≠ B ≠), a short-lived red-greenish cloud was observed to surround the nuclei. After the cloud disappeared, nuclei with yellow and mixed colors were observed moving apart from each other. The cloud was interpreted to represent the synthesis of H2B::mCherry and H2B::EGFP histones by closely located nuclei perhaps at the S phase after nuclear divisions, although the timing of H2B histone synthesis in filamentous basidiomycetes is not known. The high number of yellow nuclei in the compatible mating implies that the synthesis of histones and the packing of both histones, although not necessarily in the same amounts, into the divided nuclei occurred frequently. In the future, this interpretation has to be further certified through visualization of the nuclear divisions in the migration hyphae of different mating interactions.

In the common-A mating, with the same *A* but different *B* mating-type genes (A = B ≠), the migration hyphae contained empty compartments, red and green nuclei next to each other, but very few nuclei expressing both mCherry and EGFP. From the occurrence of red and green nuclei in the same hyphal compartment, it can be deduced that nuclear movement and septal dissolution took place, as is known to happen in matings with different *B* mating-type genes and in the haploid *B* mutant, Bon strain [2,15,16,25]. The small number of nuclei expressing both mCherry and EGFP suggests that nuclear divisions are rare in hyphae with the same *A* but different *B* mating-type genes.

In the plant pathogenic smut *Ustilago maydis*, it has been clearly shown that the binding of the pheromone to the compatible pheromone receptor induces mating of compatible haploid sporidia and increases the expression of the *a* and *b* mating-type genes [26,27]. In *S. commune* the hyphal fusions initiating the mating occur regardless of mating-type genes [3,11]. The pheromone–receptor interaction takes place after the fusion, intracellularly, and is dependent on the invading pheromone [28]. The relationship between the *A* and *B* mating-type genes has not been analyzed in tetrapolar filamentous fungi, but in the compatible mating (A ≠ B ≠), not only the number of yellow nuclei was high but also the total number of nuclei (Table 2). This indicates that the presence of both different *A* and *B* genes could enhance the intercellular nuclear migration and nuclear divisions.

The *A* mating-type genes in the A43 and A26 complexes applied here are different, and according to the published data, their products, homeodomain transcription factors, HD1 and HD2 proteins, are transcribed in the haploid and heterokaryotic migration hyphae [29,30,31,32] After hyphal fusions and the movement of a fertilizing EGFP-expressing nucleus into the mycelium with mCherry nuclei or vice versa, the nuclei with the different mating-types genes and different histones can be located next to each other. In migration hyphae, the nuclear colors change from red and green to a mixed EGFP/mCherry color, indicating that the nuclei have divided in close proximity, which is suggested to allow the accumulation of the different H2B fluorescence into the same nucleus. In the cytoplasm surrounding the nuclei, the HD1 and HD2 proteins encoded by the different *A* mating-type genes are present. These may heterodimerize and be imported into each nucleus of a closely located nuclear pair, perhaps at the same time with the newly synthesized histones, or later.

The heterodimerization of a HD1 with a HD2, each from a different *A* mating-type gene, creates an active transcription factor for the maintaining of nuclear pairing, regulation of synchronous nuclear division and clamp cell development [33,34,35,36]. In the present work, the function of the activated A-mating-type pathway in the multinucleate migration hyphae is indicated by the occurrence of nuclear pairs and incomplete clamp cells due to the expression of *A*-regulated *pcc1* and *clip1* genes [4]. The establishment of a dikaryotic mycelium is promoted at the mated colony periphery by branch formation from the migration hyphae. This is followed by the movement of a nuclear pair, each with different mating-type genes into a branch. In the branch, the synchronous nuclear division of a compatible nuclear pair, with the formation of the clamp cell and the septa [10], leads to the development of dikaryotic tip cells, and gradually to total dikaryotization of the colony edge. The involvement of branching in the development of dikaryotic hyphae has been suggested earlier [37], and a nuclear segregation event via the hyphal branches was recently suggested to play a role in the pseudosexual reproduction of the basidiomycete *Cryptococcus neoformans* [38].

## 5. Conclusions

The results obtained from the visualization of nuclei with H2B labeled with mCherry or EGFP in different *S. commune* mating interactions by spinning disc confocal microscopy emphasizes the role of cell cycle activation by the *A*-regulated pathway. In the dikaryotization process, the pairing of nuclei with different *A* and *B* mating-type genes in migration hyphae and the movement into the branches developing from the migration hyphae play a central role (Figure 6). The nuclear movements and divisions are an essential feature of the whole mating process. Research on cytoskeletal components, together with labeled nuclei in living hyphae, with the help of spinning disc confocal microscopy, could improve further our understanding of the mechanism of nuclear pairing and migration as well as the way the mechanism is regulated by mating-type genes.

## Figures and Tables

**Figure 1 jof-09-01043-f001:**
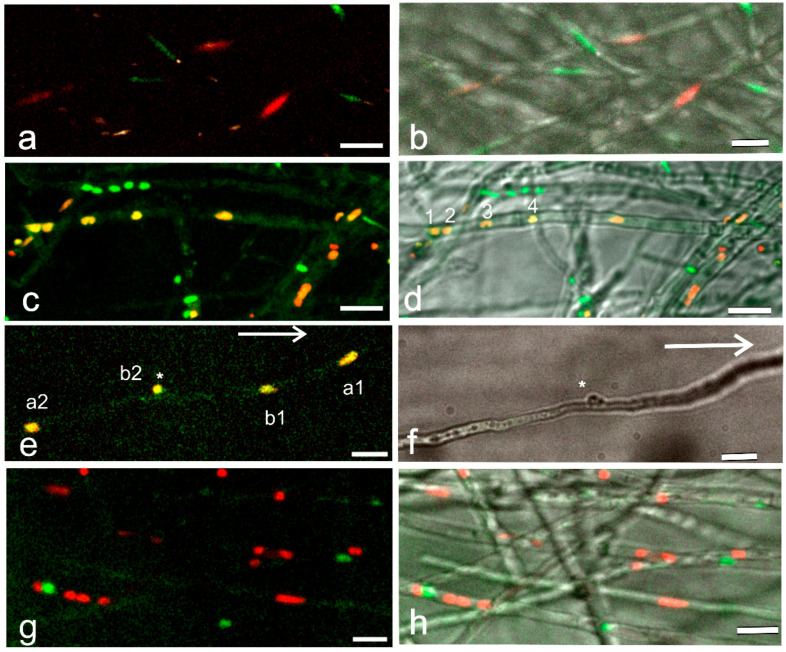
Nuclear composition in different mating interactions. Filamentous basidiomycete *Schizophyllum commune* with one of the mated strains expressing H2B::mCherry (red) and the other H2B::EGFP (green). (**a**,**c**,**e**,**g**) Projections with two colors and (**b**,**d**,**f**,**h**) with added bright field (BF) images. (**a**,**b**) Green and red nuclei in a mating between strains with the same *A* and *B* mating-type genes (A = B =). (**c**,**d**) A compatible mating between strains with different *A* and *B* mating-type genes (A ≠ B ≠). In addition to the red and green nuclei, yellow and orange ones are distinguished due to the incorporation of H2B::mCherry and H2B::EGFP in the same nucleus. In (**d**), numbers from 1 to 4 indicate the yellow nuclei analyzed in Figure 2. (**e**) Nuclei separating after conjugate division. From the nuclear division in the hypha nucleus a1 migrates toward the hyphal tip and a2 toward subapical cell. From the nuclear division at the base of the developing clamp cell (star), nucleus b1 moves toward tip cell behind a1, b2 remains for a while in the clamp cell (star). The direction of tip cell growth indicated by a long arrow. (**f**) BF image of the developing clamp cell (star). (**g**,**h**) Only red and green nuclei are distinguished in the mating with the same *A* but different *B* mating-type genes (A = B ≠). (**a**,**c**,**e**,**g**) maximum intensity Z-projections of a selected area and a number of slices at 1.25 µm intervals from the original stack and (**b**,**d**,**f**,**h**) with BF. Scale bar 10 µm.

**Figure 2 jof-09-01043-f002:**
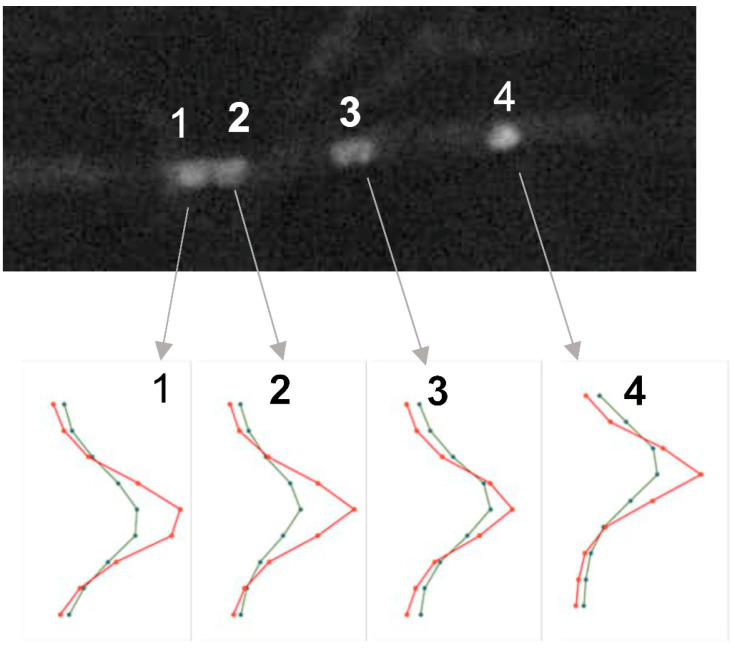
The Z-axis profile plot of the green and red color channels of the four yellow nuclei in a row in Figure 1c and with numbers in Figure 1d. The red and green color profiles line tightly indicating that red and green are present in the same nucleus.

**Figure 3 jof-09-01043-f003:**
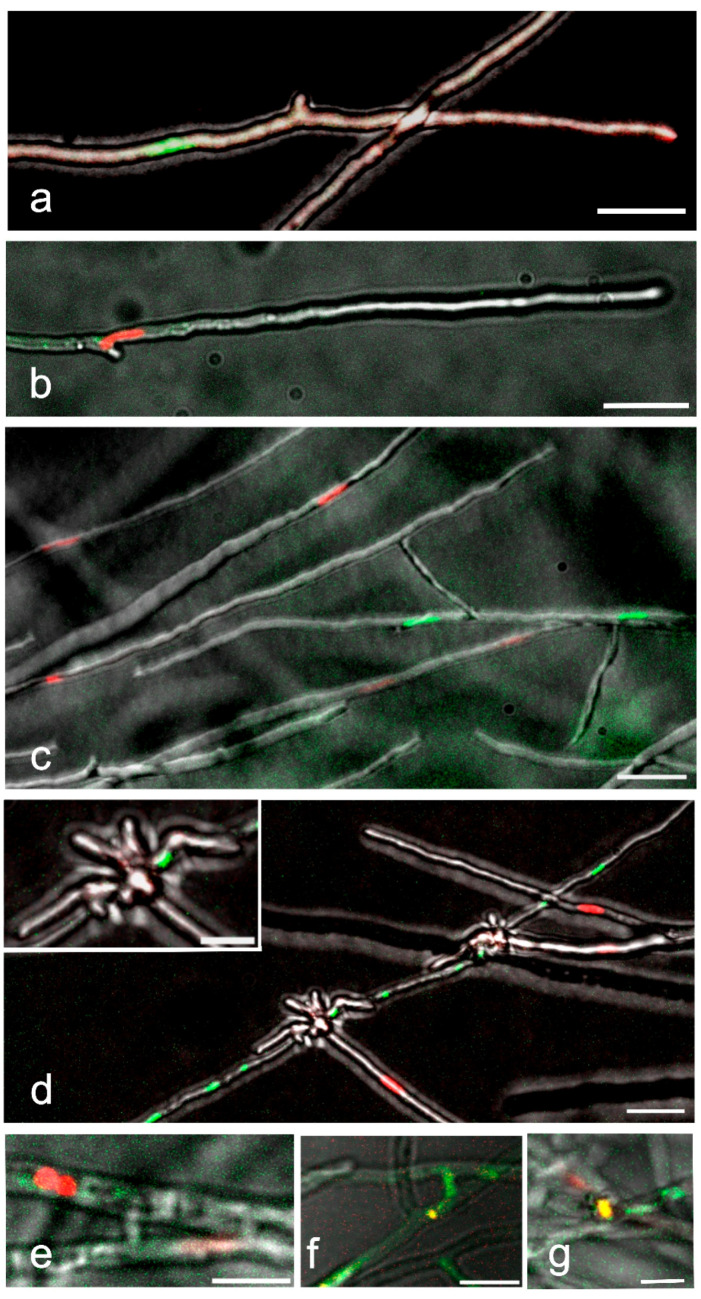
Hyphal and nuclear structure before and at the early stage of mating. (**a**,**b**). Haploid tip cells with H2B::EGFP (green) and H2B::mCherry (red) expressing nuclei and different mating-type genes. (**c**) Haploid hyphae from the mates with different mating-type genes and colored nuclei growing past each other at the early stage of confrontation. (**d**) From the same mating, a clash of hyphae with red and green nuclei. Note the branching of the clashing tips. Branches at higher magnification in the insert suggests that the nuclei may move into the branches. (**e**–**g**) Hyphal fusions (anastomoses) after the hyphae have grown deeper into the colony of the opposite mate. (**e**) Bridges between hyphae with red nuclei and (**f**) a green and yellow nucleus moving through a hyphal bridge. (**g**) A yellow nucleus at hyphal fusion from a mating with the same *A* but different *B* mating-type genes (A = B ≠). (**a**,**b**,**d**,**e**,**f**) are single-image planes, all with two colors and bright field. (**c**) the enlargement in (**d**,**g**) are maximum intensity Z-projections of a selected area and a number of slices at 1.25 µm intervals. Scale bar 20 µm in (**a**–**d**) and 10 µm in the enlargement in (**d**–**g**).

**Figure 4 jof-09-01043-f004:**
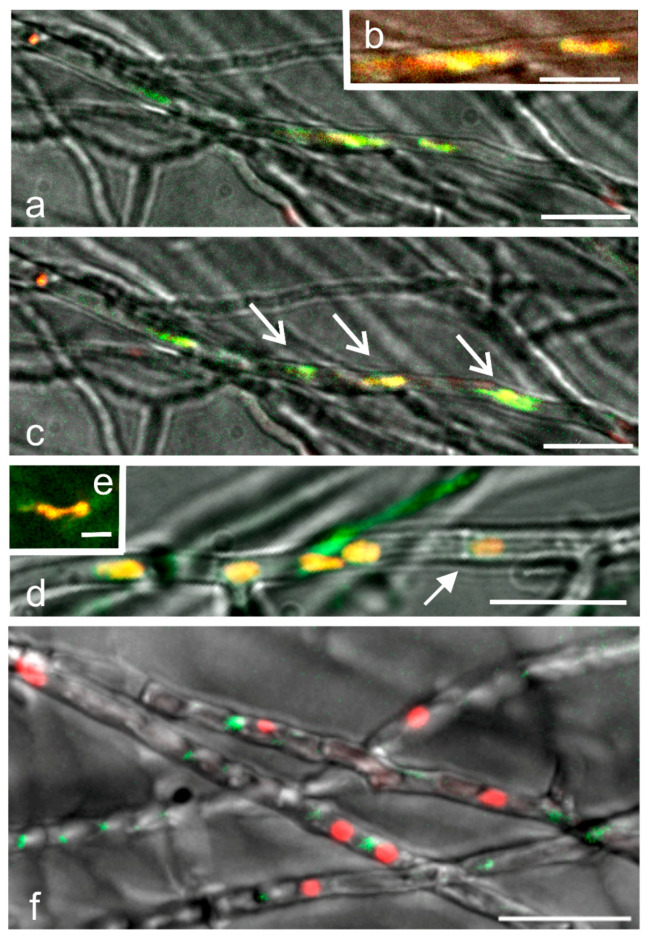
Nuclear structure in migration hyphae. (**a**–**e**) compatible (A ≠ B ≠) and (**f**) common-A (A = B ≠) mating. (**a**) A cloud of H2B::mCherry and H2B::EGFP around nuclei. (**b**) Magnification of the histone clouds in (**a**). (**c**). The fading of the histone cloud in 2 min reveals four nuclei (arrows), in the front a green and yellow nuclear pair tightly appressed, then an orange and a green nucleus, all observed to move forward (Appendix A). (**d**) At a later stage of mating, yellow and orange nuclei are common in migration hyphae. The white arrow points to an open septum with passing cytoplasmic tracks in the migration hypha. (**e**). The yellow nuclei form tight pairs, which divide synchronously. (**f**). Next to each other, red and green nuclei in the deformed hyphae of the mating with the same *A* but different *B* genes. In (**a**,**b**) a selected area from an image plane and (**c**) from Appendix A, (**d**,**e**) a selected area from a 63× image plane. (**f**). Z-projection of a 17.5 µm image stack (14 slices × 1.25 µm). All images with two colors and bright field, scale bar 20 µm, except for in (**b**) 10 and (**e**) 5 µm.

**Figure 5 jof-09-01043-f005:**
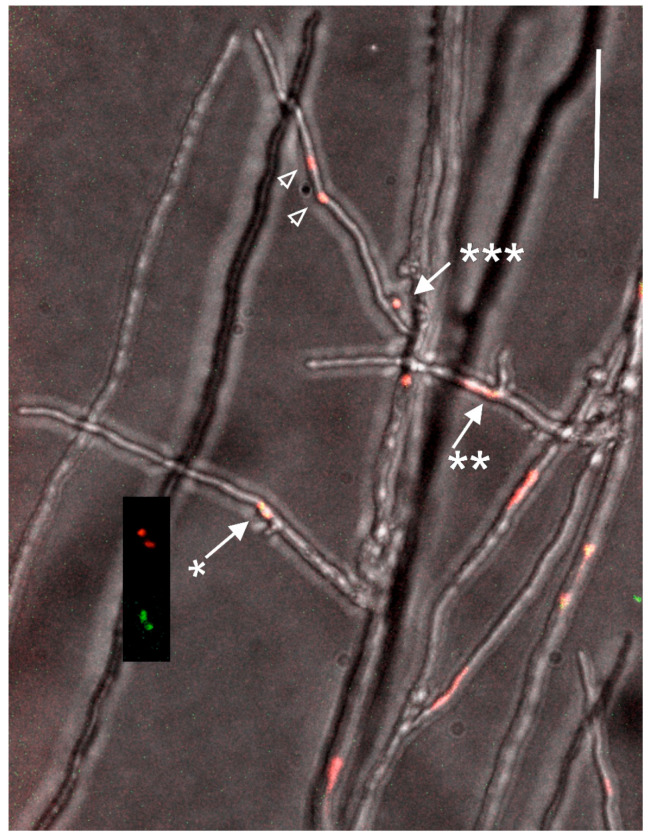
The edge of the colony during the dikaryotization process. At left, a dividing nucleus (arrow with one star) and at right (arrow with two stars) two nuclei occur in a branch. At both site a new branch develops at the site of the nuclear location. At topmost, a tip cell with two nuclei (arrow heads) and a clamp cell with an enclosed nucleus (arrow with three stars) are seen indicating that the conjugate division has taken place and the tip cell is dikaryotic. The fourth nucleus from the conjugate division has moved back to the migration hypha. All the nuclei in the figure contain histone 2B labeled with mCherry and EGFP as indicated in the insert next to the branch with the single dividing nucleus (arrow *). A single-image plane with red, green and bright field. Scale bar 50 µm.

**Figure 6 jof-09-01043-f006:**
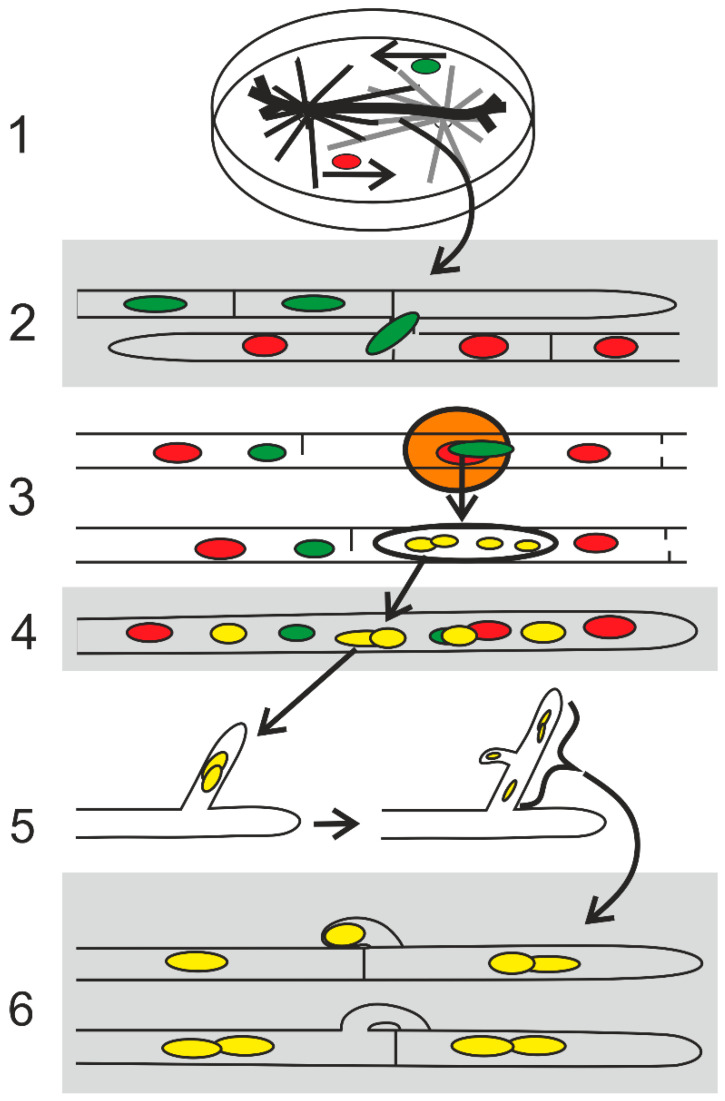
Hypothetical proposition of the nuclear behavior during the compatible mating. The different stages are based on the microscopical recordings in Figure 1, Figure 2, Figure 3, Figure 4 and Figure 5. Strains with compatible *A* and *B* genes (A ≠ B ≠) and with one strain expressing H2B::EGFP (green) and the other H2B::mCherry (red) nuclei are seen in Figure 3a,b. (**1**). During mating in a petri dish, the hyphal fusions and the movement of nuclei may occur at the confrontation line but rather between hyphae grown into the opposite colony (thick black line), Figure 3e–g. (**2**). The hyphal fusions are independent of the mating-type genes, but the breakdown of septa and nuclear movement leading to multinucleate migration hyphae is regulated by the different *B* mating-type genes (Figure 1g,h and Figure 4f). (**3**). Due to the presence of different *A* mating-type genes, the nuclear divisions are activated. The nuclei located by chance in close proximity have different *A* and *B* mating-type genes, and the concurrent synthesis of H2B::EGFP and H2B::mCherry leads to an orange cloud (halo, haze) around nuclei with different mating types (Figure 4a,b). The constitutively expressed HD transcription factors encoded by the different *A* genes in the migration hypha might heterodimerize and be imported into each of a nuclear pair, perhaps at the same time with the newly synthesized histones leading to the activation of the A pathway in both nuclei and to their synchronized nuclear division (Figure 4d,e). (**4**). A yellow nuclear pair with H2B::mCherry and H2B::EGFP and different A and B mating-type genes in each nucleus sticks together and moves into a branch from a migration hypha (arrows, Figure 5). (**5**). In the branch, the synchronous nuclear division with clamp cell development takes place (Figure 5). (**6**). The multiplicity of the phenomenon, represented here by an example of one nuclear pair, leads to complete dikaryotization of the colony edge.

**Table 1 jof-09-01043-t001:** *Schizophyllum commune* strains used in the present study.

Strains	Genotype	Interaction	Literature
I-II H2B::EGFP	*A26Bα4-β1 ura^-^h2B::egfp phleom+*		[9,10]
F16 H2B::EGFP	*A43B* *α* *4-* *β* *1 h2B::egfp phleom+*		[9,10]
4 H2B::EGFP	*A43B* *α* *3-* *β* *6 h2B::egfp phleom+*		[9,10]
T14-3-7 H2B::mCherry	*A43BB* *α* *3-* *β* *6 h2B::mCherry phloem+*		This study
**Mating interactions**			
4 H2B::EGFP × T14-3-7 H2B::mCherry	*A43Bα3-β6 h2B::egfp* × *A43BBα3-β6 h2B::mCherry*	Incompatible A = B =	This study
I-II H2B::EGFP × T14-3-7 H2B::mCherry	*A26Bα4-β1 ura-h2B::egfp* × *A43Bα3-β6 h2B::mCherry*	Compatible A ≠ B ≠	This study
F16 H2B::EGFP × T14-3-7 H2B::mCherry	*A43Bα4-β1 h2B*::egfp × *A43BBα3-β6 h2B::mCherry*	Hemicompatible A = B ≠	This study

## Data Availability

All data are publicly available.

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
