# Peer review of "Nuclear-Localized Fluorescent Proteins Enable Visualization of Nuclear Behavior in the Basidiomycete Schizophyllum commune Early Mating Interactions"

_jof, 2023, doi:10.3390/jof9111043_

Round 1

Reviewer 1 Report

This is a fundamental study on fungal mating reactions, which is interesting and important to understand the nuclear interaction process of compatible hyphae in the early stages of mating reactions.

A slight drawback is that the author uses a spinning disc confocal microscope. Although it is ideal for shooting speed, the imaging quality is slightly poor, which may be related to the lens quality. In addition, if routine nuclear staining and imaging were performed simultaneously, it would be more perfect.

Here are some details and questions,

In the title, it should be reflected that this is the nuclear interaction behavior in the early stage of mating reaction.

Please clarify whether the F16 H2B:: EGFP strain is referenced or constructed in this article, and the material method (line88-line108) is inconsistent with the description in Table 1.

Line 135, 232, 276,”1,25” should be “1.25”.

In Table 2, in the fully compatible reaction, the different nuclear color types and total numbers obtained are inconsistent.

In the so-called incompatible pairing (A=B=), there should be a certain amount of hyphal-self fusion. Only one yellow nucleus was observed in this article. Is this related to the early stage of affinity response in the area detected by the author? This issue can be discussed in an appropriate location. Correspondingly, in compatible reactions, the presence of nuclear localization models should result in all or a large number of nuclei in the cell presenting a bicolor state, especially in mature cells.

The position of the septum needs to be identified in the key images, as they cannot be observed in existing bright field photos.

In Figure 6, the fourth process is puzzling. How does this multinucleated cell state occur and is there sufficient evidence to prove it?

Author Response

Response to Reviewer 1

Comments and Suggestions for Authors

This is a fundamental study on fungal mating reactions, which is interesting and important to understand the nuclear interaction process of compatible hyphae in the early stages of mating reactions.

A slight drawback is that the author uses a spinning disc confocal microscope. Although it is ideal for shooting speed, the imaging quality is slightly poor, which may be related to the lens quality. In addition, if routine nuclear staining and imaging were performed simultaneously, it would be more perfect.

I am sorry about the low imaging quality. We should have worked more with 100X magnification. With the thick samples used, the spinning disk confocal microscopy provided the necessary information faster and more easily than the confocal microscope that I have previously used in the study of dikaryotic mycelium (ref.10).

With routine DAPI/Hoechst nuclear staining 1) it would not have been possible to observe living hyphae in Schizophyllum commune and 2) it would have been impossible to recognize the nuclei containing the different mating-types, as it is was now possible at the early phase of the compatible (A≠B≠), hemicompatible (A=B≠) and incompatible (A=B=) matings.

Here are some details and questions,

In the title, it should be reflected that this is the nuclear interaction behavior in the early stage of mating reaction.

The word early has been included in the title.

Please clarify whether the F16 H2B:: EGFP strain is referenced or constructed in this article, and the material method (line88-line108) is inconsistent with the description in Table 1.

I am sorry, strain 16 was constructed for the reference 9, corrected in Table 1.

Line 135, 232, 276,”1,25” should be “1.25”.

Corrected in the manuscript.

In Table 2, in the fully compatible reaction, the different nuclear color types and total numbers obtained are inconsistent.

The number is 232, not 229, sorry.

1)In the so-called incompatible pairing (A=B=), there should be a certain amount of hyphal-self fusion. Only one yellow nucleus was observed in this article. Is this related to the early stage of affinity response in the area detected by the author? This issue can be discussed in an appropriate location. 2)Correspondingly, in compatible reactions, the presence of nuclear localization models should result in all or a large number of nuclei in the cell presenting a bicolor state, especially in mature cells.

1) I assume that the one yellow nucleus in (A=B=) is a consequence of the division of a green and a red nucleus with same mating type in a fusion bridge between the strains, which could lead to the incorporation of both green and red histones in the same nucleus. As mentioned in the manuscript, the hyphal fusions are independent of mating-type genes. This may happen on very rare occasions. The same situation is seen in Fig. 3g, in the common A mating (A=B≠). This has to be studied further in the future.

2) Fig. 1c and d present a fully developed migration hyphae of a compatible mating. Green, red and yellow nuclei are seen.

The position of the septum needs to be identified in the key images, as they cannot be observed in existing bright field photos.

There are not many septa to identify. In compatible and common A matings, (A=B≠)  the septa are broken down due to the activity of B mating-type genes. In Fig. 3a-d, the monokaryotic hyphae and their confrontation describe the region around the nucleus and the tip of the apical cells. The apical cells can be 150 to 200 µm long, the nucleus is in the middle of the apical cell (ref.10) and the septum is about 75 to100 µm behind the nucleus, not seen in the Figs. In Fig. 4, the broken septum has been marked in the migration hypha of a compatible mating.

In Figure 6, the fourth process is puzzling. How does this multinucleated cell state occur and is there sufficient evidence to prove it.

The evidence for phase 4 in Fig. 6 comes from the following microscopic figures: Fig. 1 c, d, e, f. Fig. 3. d and f, and video 2. In the common A mating-type (A=B≠), Fig. 1 g and h, Fig.4 h, the red and green nuclei are next to each other and in the compatible (A≠B≠) mating, Fig. 1 c and d, Fig. 4d, the yellow nuclei are present in addition to the red and green ones already seen in the common A (A=B≠) mating. The microscopic sources for the different phases in the schematic drawing have been added to the Fig. 6 text.

Submission Date

12 September 2023

Date of this review

27 Sep 2023 08:58:09

 Author Report Rating(Optional)

 Review Report rating refer to the guideline: https://www.mdpi.com/reviewers#Review_Report

Reviewer 2 Report

First, before discussing the role of the A and B factors in the mating process and during dikaryotic growth, it would be good to describe in more detail the process of nuclear migration and the establishment of a dikaryon in S. commune. In the present setup the reader is submersed immediately in the details of the molecular regulation.

Second, it would be useful to introduce the phenotypes of common A and common B heterokaryons for S. commune in the introduction, and to formulate hypotheses of the nuclear behaviour and coloration of the common A heterokaryon.

Third, it would be useful to have in the introduction a schematic figure of the known and unknown events after a mating between compatible and common-A and common-B matings. Alternatively, and perhaps preferably, such a figure could be provided at the start of the results section, including reference to the various figures in the remainder of the results section.

Finally, I think your main finding can be made more clear, particularly in the abstract. In the conclusions section, the finding that in compatible, but not incompatible or hemicompatible matings, a large number of yellow nuclei is found, and that you interpret this as nuclear division, is not mentioned. Please clarify this and align the text of the abstract and conclusions section.

Detailed remarks:

The writing is unnecessarily complex and indirect. For example, you can define early on your mutant strains with a more simple code (for example “green” or “red”). Another example: In the conclusion section, I expect a brief statement of the main conclusions, and not a sentence like the first one “The results obtained from the visualization of nuclei with H2B labeled with mCherry 392 or EGFP in S. commune mating interactions by spinning disc confocal microscopy can be interpreted based on the existing knowledge of interactions of pheromones and receptors encoded by B and homeodomain transcription factors encoded by A mating type genes.”. Please remove this sentence.

Instead of hemicompatible (A=B≠) or only hemicompatible, I would recommend the use of common-A mating, as this is a well-established term and describes the mating exactly.

Am I correct that strains 4 H2B and T14-3-7 are isogenic? Since these strains carry identical A and B alleles, they are incompatible, but I was wondering if there was any other incompatibility precluding successful hyphal fusion.

First sentence of the results section: Please add “incompatible” to “mating (A=B=)”.

In the conclusions section: “the nuclear segregation and movement into the branches developing from the migration hyphae plays a central role” please change “plays” to “play”

Okayish, can be improved, particularly more direct language and simplification of terms (see detailed comments).

Author Response

Comments and Suggestions for Authors

First, before discussing the role of the A and B factors in the mating process and during dikaryotic growth, it would be good to describe in more detail the process of nuclear migration and the establishment of a dikaryon in S. commune. In the present setup the reader is submersed immediately in the details of the molecular regulation.

Second, it would be useful to introduce the phenotypes of common A and common B heterokaryons for S. commune in the introduction, and to formulate hypotheses of the nuclear behaviour and coloration of the common A heterokaryon.

As an answer to the two comments above I have added the following text to the end of Introduction

In the confronted haploid strains, the different B mating-type genes induce the reciprocal exchange and migration of nuclei as well as the breakdown of the septa in the interacting strains [2]. This is seen as a “flat” phenotype, not recognized in other filamentous basidiomycetes. Microscopically, the ”flat” phenotype has mainly been studied in the haploid B-mutant strain, which is equivalent to a mating with different B genes (A=B≠) [15, 16], and to a lesser extent in the confrontation of haploid strains with common A but different B genes, as in the present study using spinning disc confocal microscopy. A compatible mating is usually recognized by the formation of clamp connections at the colony edge. No information is available about the A regulated process that organizes the nuclei into synchronously dividing nuclear pairs with different mating type genes. Analyses of the migration hyphae of a compatible mating, in which the nuclei with different mating-type genes are recognized due to their fluorescent colours, revealed that nuclear divisions play a central role in the dikaryotizaton process.

Third, it would be useful to have in the introduction a schematic figure of the known and unknown events after a mating between compatible and common-A and common-B matings. Alternatively, and perhaps preferably, such a figure could be provided at the start of the results section, including reference to the various figures in the remainder of the results section.

I hope that the schematic presentation (Fig. 6) is sufficient. In the Figure 6 text I have included references to the Figures in Result section. I would be very grateful if this arrangement is satisfactory at the present stage.

Finally, I think your main finding can be made more clear, particularly in the abstract. In the conclusions section, the finding that in compatible, but not incompatible or hemicompatible matings, a large number of yellow nuclei is found, and that you interpret this as nuclear division, is not mentioned. Please clarify this and align the text of the abstract and conclusions section.

Below the improved Abstract

Abstract: Spinning disc confocal microscopical research was conducted on living mating hyphae of the tetrapolar basidiomycete Schizophyllum commune. Haploid strains with either the same or different A and B mating-type genes and expressing differently labelled histone 2B were confronted. In the haploid hyphae histone 2B mCherry and histone 2B EGFP were visualized as red and green nuclei, respectively. In hyphae with the same A but different B genes, the red and green nuclei were observed next to each other. This indicated that nuclear migration between strains, regulated by the B mating-type, had taken place. The compatible mating with different A and B genes produced a high number of mixed EFGP/mCherry nuclei. The mixed nuclei resulted from nearby divisions of nuclei encoding different histones and mating-type genes. During this process, the histones with the different labels were incorporated in the same nuclei, along with the heterodimerized transcription factors encoded by the different A mating-type genes and present around the nuclei. This led to the activation of the A-regulated pathway and indicated that different A genes are important to the cell cycle activation of a compatible mating. Consequently, a yellow nuclear pair stuck together, divided synchronously and proceeded in the migration hyphae towards the colony periphery, where the dikaryotization was promoted by branch formation from the migration hyphae.

Detailed remarks:

The writing is unnecessarily complex and indirect. For example, you can define early on your mutant strains with a more simple code (for example “green” or “red”). Another example: In the conclusion section, I expect a brief statement of the main conclusions, and not a sentence like the first one “The results obtained from the visualization of nuclei with H2B labeled with mCherry 392 or EGFP in S. commune mating interactions by spinning disc confocal microscopy can be interpreted based on the existing knowledge of interactions of pheromones and receptors encoded by B and homeodomain transcription factors encoded by A mating type genes.”. Please remove this sentence.

The sentence from the beginning of Conclusions has been removed.

Instead of hemicompatible (A=B≠) or only hemicompatible, I would recommend the use of common-A mating, as this is a well-established term and describes the mating exactly.

The term Common-A mating has been included/used instead of or in addition to hemicompatible.

Am I correct that strains 4 H2B and T14-3-7 are isogenic? Since these strains carry identical A and B alleles, they are incompatible, but I was wondering if there was any other incompatibility precluding successful hyphal fusion.

They are isogenic for their mating-type genes. But the strain 4 H2B was obtained from the crossing I-11::H2B(A26(Bα1-β4 EGFP )X A43Bα3-β6 in the previous work (ref. 9). The strain I-11::H2B  used in the original transformation (ref. 9) has not been sequenced. It cannot be said that the strains are isogenic. For the second question I do not have an answer.

First sentence of the results section: Please add “incompatible” to “mating (A=B=)”.

Added

In the conclusions section: “the nuclear segregation and movement into the branches developing from the migration hyphae plays a central role” please change “plays” to “play”

Corrected.